# Dietary Calcium and Protein Levels Influence Growth Performance, Intestinal Development, and Nutrient Utilization in Goslings

**DOI:** 10.3390/vetsci12040310

**Published:** 2025-03-28

**Authors:** Yuanjing Chen, Guoqiang Su, Ning Li, Zhengfeng Yang, Haiming Yang, Zhiyue Wang

**Affiliations:** 1College of Veterinary Medicine, Yangzhou University, Yangzhou 225000, China; mx120170656@yzu.edu.cn; 2College of Animal Science and Technology, Yangzhou University, Yangzhou 225000, China; quentinsugq@outlook.com (G.S.); mz120191051@stu.yzu.edu.cn (N.L.); 2023019@ntu.edu.cn (Z.Y.); hmyang@yzu.edu.cn (H.Y.)

**Keywords:** calcium, goslings, growth performance, intestinal development, nutrient utilization, protein

## Abstract

Proper nutrition is essential for the healthy growth and development of goslings, particularly during the early brooding period. Calcium (Ca) and crude protein (CP) are two fundamental dietary components that influence skeletal development, metabolism, and overall growth performance. However, the precise dietary requirements for goslings remain insufficiently studied. This study aimed to investigate how different levels of dietary Ca and CP affect growth, nutrient utilization, intestinal morphology, and digestive enzyme activities in Jiangnan White goslings. A 3 × 3 factorial design was implemented, testing three levels of Ca (0.32%, 0.96%, and 2.88%) and three levels of CP (14.5%, 18.5%, and 22.5%). Results indicate that moderate levels of both Ca (0.96%) and CP (14.5–18.5%) optimize growth, nutrient absorption, and intestinal health, enhancing body weight, average daily gain, and feed intake. Excessive Ca (2.88%) negatively impacted growth and feed conversion efficiency. Intestinal morphology and digestive enzyme activities were improved with moderate nutrient levels. The findings highlight the importance of balanced dietary formulations for optimizing growth performance and metabolic efficiency in goslings, offering valuable insights for poultry production.

## 1. Introduction

Optimal growth, development, and health in goslings depend on a nutritionally balanced diet, with calcium and crude protein being fundamental nutrients that regulate skeletal development, metabolism, and overall physiological function. Protein serves as the primary building block for tissue formation, enzymatic activity, and metabolic regulation, ensuring proper growth and physiological homeostasis [1]. Calcium, one of the most abundant minerals in the body, plays a critical role in bone mineralization, neuromuscular function, and cellular signaling pathways [2]. However, imbalances in dietary calcium and protein—whether deficiency or excess—can impair nutrient metabolism, disrupt physiological processes, and lead to metabolic disorders that negatively affect growth performance [3,4].

Although poultry nutrition research has extensively studied nutrient requirements for broilers and layers, relatively fewer studies focus on goslings, despite their unique nutrient metabolism and growth characteristics. The National Research Council (NRC, 1994) recommends a dietary calcium range of 0.6–1.0% for growing waterfowl, while more recent studies suggest that goslings may require slightly higher levels (~0.9%) due to increased growth rates and bone mineralization demands [5]. Similarly, the optimal crude protein concentration for goslings is reported to range between 16% and 20%, supporting muscle development, enzymatic activity, and overall metabolic efficiency. The NRC (1994) [5] recommends a protein intake of 16–18% for growing goslings, but some recent studies indicate that higher dietary protein levels (≥22%) may not further enhance growth and could increase nitrogen excretion, potentially leading to metabolic inefficiencies and environmental concerns [6]. These findings highlight the importance of balancing protein intake to maximize growth while minimizing waste nitrogen output.

The interaction between dietary calcium and protein levels influences nutrient absorption, utilization, and overall metabolic efficiency [7]. Excess dietary calcium can interfere with protein digestion and amino acid absorption by forming insoluble complexes with phosphorus and proteins, reducing their bioavailability in the intestine. Conversely, inadequate protein intake may impair calcium metabolism by affecting calcium-binding proteins, intestinal absorption, and renal excretion rates. Additionally, an improper calcium-to-phosphorus ratio can impair bone mineralization, alter enzymatic activities, and affect feed conversion efficiency, emphasizing the need for a well-balanced diet. Understanding these nutrient interactions is critical for optimizing dietary formulations, improving nutrient retention, and mitigating metabolic complications in goslings.

Goslings are particularly vulnerable to nutritional metabolic diseases resulting from inappropriate calcium and protein levels [8]. Excess protein intake can contribute to avian gout due to increased uric acid production and renal overload, while excessive dietary calcium can lead to hypercalcemia, which disrupts phosphorus metabolism and contributes to soft tissue mineralization and nephrocalcinosis. Calcium deficiency or an imbalanced calcium-to-phosphorus ratio may result in rickets and skeletal deformities, leading to poor bone development and increased susceptibility to fractures. Given the rapid growth phase during the brooding period, goslings have heightened nutritional requirements, making dietary optimization essential for maximizing production efficiency and preventing diet-induced metabolic disorders [9]. Despite their importance, precise dietary recommendations for calcium and protein levels in goslings remain limited, particularly under modern production systems.

The present study aims to investigate the effects of dietary calcium and protein levels on the growth performance, intestinal development, and nutrient utilization of Jiangnan white goslings during the brooding period. Furthermore, it aims to establish a scientific foundation for reducing the prevalence of nutritional metabolic diseases and enhancing production efficiency.

## 2. Materials and Methods

### 2.1. Ethics Statement

This study was subjected to rigorous review and subsequently approved by the Institutional Animal Care and Use Committee at Yangzhou University (approval number: SYXK [Su] 2020-0910).

### 2.2. Animals and Dietary Treatments

A total of 972 one-day-old male Jiangnan White goslings (average body weight: 99 ± 5 g), sourced from Jiangsu LIHUA Animal Husbandry Co., were used in a 30-day feeding trial. The experiment employed a two-factor factorial design (3 × 3) to investigate the effects of dietary calcium (Ca) and crude protein (CP) levels on gosling growth and development. The study comprised nine dietary treatments, with three levels of calcium (0.32%, 0.96%, 2.88%) and three levels of crude protein (14.5%, 18.5%, 22.5%), as shown in Table 1. Each treatment group included six replicates, with 18 goslings per replicate.

The experimental diets were formulated as corn–soybean meal-based powdered feeds, designed to meet the nutritional requirements of Jiangnan White goslings. We employed a comprehensive ingredient adjustment strategy to maintain an optimal nutritional balance. This approach was necessary to ensure that variations in Ca and CP levels did not cause unintended deficiencies or excesses in other essential nutrients such as energy, amino acids, fiber, and trace minerals. Low levels (0.32% Ca, 14.5% CP) represent potential deficiencies, medium levels (0.96% Ca, 18.5% CP) align with standard feeding recommendations, and high levels (2.88% Ca, 22.5% CP) assess the effects of nutrient excess. Nutrient levels were established based on recommendations from the National Research Council (NRC, 1994) [5], the Chinese Feed Composition and Nutritional Value Table (30th edition, 2019), and prior research conducted in our laboratory specific to this breed. The detailed composition and nutritional profile of the basal diet are presented in Table 2.

### 2.3. Bird Management

The experiment was conducted at the Modern Agricultural Science and Education Demonstration Park of Yangzhou University. The goslings were housed in wire-floor pens measuring 1.9 m × 1.5 m, located in a controlled environment with an ambient temperature maintained at 28.5 °C (range: 26.5–30.5 °C). Water and feed were provided ad libitum throughout the experimental period. The pens were regularly cleaned to maintain hygiene and ensure proper ventilation.

For lighting management, goslings were exposed to 24 h of light per day from day 1 to day 3 of age. Subsequently, the light duration was gradually reduced by 1 h per day until it aligned with natural daylight conditions.

### 2.4. Sample Collection and Analytical Determination

On days 14 and 30 of the experiment, one bird was randomly selected from each replicate for sampling. The selected birds were weighed and humanely slaughtered via jugular exsanguination. Blood samples were collected aseptically from the wing vein using sterile needles and syringes. The collected blood was allowed to clot at room temperature and subsequently centrifuged at 3000 rpm for 10 min to separate the serum, which was then stored for further biochemical analysis.

For morphological evaluation, 1 cm sections of the duodenum, jejunum, and ileum were excised and immediately fixed in 4% paraformaldehyde. Additionally, the contents of the duodenum, jejunum, and ileum were carefully collected, snap-frozen in liquid nitrogen, and stored at −20 °C for subsequent analyses of enzyme activity.

#### 2.4.1. Growth Performance

Upon arrival at the experimental farm from the hatchery, all goslings were individually weighed to document their initial body weight. At 30 days of age, body weight (BW), average daily feed intake (ADFI), average daily gain (ADG), and feed conversion ratio (FCR) were recorded following a 6 h period of feed deprivation to ensure consistency in measurements.

#### 2.4.2. Serum Parameters

Serum uric acid (UA), creatinine (Cr), urea nitrogen (UN), calcium (Ca), and phosphorus (P) concentrations were determined using commercial assay kits, following the manufacturer’s protocols (Nanjing Jiancheng Bioengineering Institute, Nanjing, Jiangsu Province, China).

#### 2.4.3. Intestinal Morphology

The preparation of intestinal tissue sections was conducted following the protocol described by Yang et al. [10]. Villus height (VH) and crypt depth (CD) were measured at 400× magnification using a universal video imaging system (LY-WN-HP SUPER CCD) manufactured by Chengdu Liyang Precision Electromechanical Co., Ltd. (Chengdu, China) VH was measured from the tip of the villus to the villus–crypt junction, while CD was measured from the base of the villus to the muscularis mucosa. Measurements were averaged from six randomly selected sections per sample to ensure reliable data representation.

#### 2.4.4. Assessment of Intestinal Digestive Enzyme Activity

The levels of α-amylase (α-AMS), lipase (LPS), and trypsin (TPS) in the chyme from each intestinal segment were quantified using standardized enzymatic assays. The specific methodologies employed were as follows: (1) α-AMS activity was determined using the starch-iodine colorimetric method; (2) LPS activity was measured using the methyl resorufin substrate method; and (3) TPS activity was assessed using the N-benzoyl-L-arginine ethyl ester (BAEE) method. For each intestinal segment (duodenum, jejunum, ileum), approximately 0.5 g of intestinal content per replicate was used for digestive enzyme activity assays. Prior to analysis, frozen samples were thawed on ice and homogenized in nine volumes of cold physiological saline solution (0.9% NaCl, *w*/*v*) using a tissue homogenizer at 4 °C. The homogenate was then centrifuged at 12,000× *g* for 10 min at 4 °C, and the resulting supernatant was collected for enzyme activity determination. All assay kits were obtained from Nanjing Jiancheng Bioengineering Institute, Jiangsu Province, China, and the procedures were conducted according to the manufacturer’s protocols.

#### 2.4.5. Apparent Utilization of Calcium, Phosphorus, and Nitrogen

The metabolic experiment commenced when the goslings reached 20 days of age. Six goslings were selected per treatment group, and fecal samples were collected using the total feces collection method. A collection tray was placed beneath each pen to gather all excreted feces during the designated collection period. To prevent contamination and ensure complete recovery, the trays were lined with plastic sheets. Feces were collected daily to minimize nutrient loss or contamination and were immediately stored at −20 °C for further analysis. The experiment consisted of a 4-day adaptation period followed by a 3-day data collection period. After all fecal samples were collected, the excreta from each gosling were thoroughly mixed, dried to a constant weight in a 65 °C oven, rehydrated to ambient moisture levels for 24 h, and ground to a fine powder using a 40-mesh sieve (0.45 mm aperture). Protein content was analyzed using the semi-micro Kjeldahl method, Ca content was determined using the MTB microplate method, and P content was measured using the phosphomolybdic acid method [11].

The apparent digestibility of calcium, phosphorus, and protein was determined using the following formula:
$$\mathrm{Apparent}\;\mathrm{Digestibility}\;(\%)=\frac{Nutrient\;intake-Fecal\;nutrient\;excretion}{Nutrient\;intake}\times 100$$

### 2.5. Statistical Analysis

Data processing and statistical analyses were performed in replicates. Experimental data were organized using Excel 2017 and analyzed with SPSS 22.0. A two-way analysis of variance (ANOVA) with interaction effects was conducted using the general linear model (GLM) procedure to evaluate inter-factor effects. Between-group comparisons were conducted using one-way ANOVA, followed by Tukey’s post hoc test for multiple comparisons. Statistical significance was defined as *p* < 0.05.

## 3. Results

### 3.1. Growth Performance

The effects of diets with varying calcium and protein levels on the growth performance of goslings from 1 to 30 days of age are presented in Table 3 and Appendix A. As shown in the table, dietary Ca levels significantly influenced BW at 30 days (*p* < 0.001). Goslings fed 0.32% or 0.96% Ca diets had significantly higher BW (1779.55 g and 1762.85 g, respectively) compared to those fed 2.88% Ca (823.57 g). CP levels also significantly affected 30-day BW (*p* = 0.001), with the highest BW observed in goslings fed 14.5% CP (1526.66 g).

ADFI was significantly affected by both Ca (*p* < 0.001) and CP levels (*p* = 0.008). The highest ADFI was observed in goslings fed 0.96% Ca (123.44 g) and 14.5% CP (105.90 g), while the lowest ADFI was recorded in those fed 2.88% Ca (63.89 g). ADG was significantly reduced in goslings fed 2.88% Ca (*p* < 0.001), with the highest ADG observed in the 0.96% Ca group (56.51 g). CP levels also influenced ADG (*p* = 0.046), with the 14.5% CP group showing the highest ADG (47.39 g). FCR was significantly higher in the 2.88% Ca group (2.74, *p* < 0.001) compared to the 0.32% and 0.96% Ca groups (2.06 and 2.19, respectively). CP levels had no significant effect on FCR (*p* = 0.507), although a significant interaction between Ca and CP was observed (*p* = 0.015).

### 3.2. Serum Parameters

Serum UA, Cr, and UN concentrations at 14 and 30 days under varying dietary Ca and CP levels are summarized in Table 4 and Appendix A. At 14 days of age, dietary calcium levels significantly influenced serum UA, Cr, and UN concentrations (*p* < 0.05). Goslings fed a diet with 2.88% Ca exhibited significantly higher UA (308.73 µmol/L) and Cr (49.46 µmol/L) levels compared to those fed diets with 0.32% Ca (203.74 µmol/L for UA, 36.91 µmol/L for Cr) or 0.96% Ca (274.31 µmol/L for UA, 20.47 µmol/L for Cr). Conversely, UN levels were significantly lower in goslings fed the 0.96% Ca diet (0.64 µmol/L) compared to those fed diets with 0.32% Ca or 2.88% Ca (*p* < 0.001).

Similarly, dietary protein levels had significant effects on serum biochemical parameters *p* < 0.001). Goslings fed a 22.5% CP diet exhibited the highest Cr (39.95 µmol/L) and UN (2.60 µmol/L) levels, whereas those fed an 18.5% CP diet had the lowest Cr (36.80 µmol/L) and UN (1.62 µmol/L) levels. Additionally, UA levels were significantly higher in goslings fed an 18.5% CP diet (281.39 µmol/L) compared to those fed a 14.5% CP diet (241.30 µmol/L, *p* = 0.005).

At 30 days of age, similar trends were observed. Goslings fed a 2.88% Ca diet exhibited significantly higher Cr (30.89 µmol/L) and UN (2.89 µmol/L) concentrations compared to those fed diets with 0.96% Ca (16.71 µmol/L for Cr, 1.08 µmol/L for UN) or 0.32% Ca (39.78 µmol/L for Cr, 3.01 µmol/L for UN, *p* < 0.001). The highest UA levels were observed in goslings fed the 0.32% Ca diet (277.22 µmol/L); however, this difference was not statistically significant (*p* = 0.052).

Dietary protein levels also significantly influenced serum parameters at 30 days of age (*p* < 0.001). Goslings fed a 22.5% CP diet exhibited the highest UA (301.71 µmol/L) and Cr (31.42 µmol/L) concentrations, while those fed an 18.5% CP diet had the lowest UN levels (1.70 µmol/L, *p* < 0.001). Furthermore, significant interaction effects between calcium and protein levels were observed for most serum parameters at both 14 and 30 days (*p* < 0.001).

Serum Ca and P concentrations at 14 and 30 days under varying dietary Ca and CP levels are summarized in Table 5 and Appendix A. At 14 days, dietary Ca significantly affected serum Ca (*p* < 0.001), with the highest level in the 0.32% Ca group (2.57 mmol/L), compared to 0.96% (2.30 mmol/L) and 2.88% Ca (2.41 mmol/L). Serum P was also higher in the 0.32% Ca group (2.52 mmol/L), but differences were not significant (*p* = 0.065). Dietary CP had no significant effect on serum Ca or P (*p* = 0.420, *p* = 0.186), and no interaction between Ca and CP was observed (*p* > 0.05).

At 30 days, dietary Ca continued to influence serum Ca (*p* = 0.027), with 2.88% Ca showing the highest levels (2.57 mmol/L). Serum P was significantly higher in the 0.32% and 2.88% Ca groups compared to 0.96% Ca (*p* < 0.001). CP had no significant effect on serum Ca (*p* = 0.257), but significantly influenced serum P (*p* = 0.019), with higher levels in the 14.5% and 18.5% CP groups (2.37 mmol/L and 2.36 mmol/L) than in the 22.5% CP group (2.23 mmol/L). No significant Ca × CP interaction was observed (*p* > 0.05).

### 3.3. Intestinal Morphology

The effects of dietary Ca and CP levels on the morphology of the duodenum, jejunum, and ileum in goslings at 14 and 30 days of age are summarized in Table 6, Table 7, Table 8 and Appendix A.

#### 3.3.1. Duodenal Morphology

At 14 days, VH and CD in the duodenum were significantly influenced by dietary Ca levels (*p* < 0.05). Goslings fed a 0.96% Ca diet exhibited the highest VH (988.42 µm) and CD (230.70 µm), whereas those on a 2.88% Ca diet showed the lowest VH (743.59 µm) and CD (201.70 µm). CP levels significantly affected VH (*p* < 0.05), with goslings fed 18.5% CP showing the highest VH.

At 30 days, dietary Ca continued to significantly affect VH and CD (*p* < 0.05). The highest VH was observed in the 0.96% Ca group, while the 2.88% Ca group had the lowest VH (750.34 µm). CP levels influenced CD (*p* < 0.05), with the highest CD (247.28 µm) recorded in goslings fed 14.5% CP. Significant interaction effects were observed between Ca and CP for CD (*p* < 0.05).

#### 3.3.2. Jejunal Morphology

At 14 days, dietary Ca significantly affected VH (*p* < 0.001) and CD (*p* = 0.021). The highest VH (1055.12 µm) and CD (242.97 µm) were observed in the 0.32% Ca group. CP levels also significantly influenced VH (*p* < 0.05), with goslings fed 22.5% CP showing the highest VH (944.02 µm).

At 30 days, dietary Ca significantly influenced VH and CD (*p* < 0.05). The 0.96% Ca group had the highest VH (1290.75 µm) and CD (250.88 µm), while the 2.88% Ca group had the lowest values (1095.55 µm for VH, 211.09 µm for CD). CP levels also affected VH and CD (*p* < 0.05), with the highest VH (1253.45 µm) and CD (250.03 µm) recorded in the 22.5% CP group. Significant interaction effects were observed for both VH and CD (*p* < 0.05).

#### 3.3.3. Ileal Morphology

At 14 days, dietary Ca levels significantly influenced VH (*p* < 0.001). The highest VH (859.32 µm) was observed in the 0.32% Ca group, whereas the 2.88% Ca group had the lowest VH (702.97 µm). CP levels also significantly affected VH (*p* = 0.005), with the 22.5% CP group showing the highest VH (798.21 µm).

At 30 days, dietary Ca significantly influenced VH (*p* < 0.001) and CD (*p* = 0.046). Goslings fed 0.96% Ca had the highest VH (1067.69 µm), while the 2.88% Ca group showed the lowest VH (829.82 µm). CP significantly affected VH (*p* = 0.026), with the highest VH (996.35 µm) observed in the 18.5% CP group. Interaction effects between Ca and CP were significant for VH and CD (*p* < 0.05).

### 3.4. Assessment of Intestinal Digestive Enzyme Activity

The effects of dietary Ca and CP levels on the digestive enzyme activities of α-AMS, LPS, and TPS in the duodenum, jejunum, and ileum of goslings at 14 and 30 days are summarized in Table 9, Table 10, Table 11 and Appendix A.

#### 3.4.1. Duodenal Digestive Enzyme Activities

At 14 days, dietary Ca levels significantly influenced α-AMS and LPS activities (*p* < 0.05). The highest α-AMS activity (0.34 U/mg prot) was observed in the 2.88% Ca group, while the 0.96% Ca group exhibited the highest LPS activity (27.81 U/mg prot). CP levels significantly affected α-AMS (*p* < 0.001) and LPS (*p* = 0.010), with the highest activities recorded in goslings fed 14.5% CP. Significant interaction effects between Ca and CP were observed for all enzymes (*p* < 0.05).

At 30 days, α-AMS and TPS activities were significantly higher in the 0.32% Ca group (*p* < 0.001). CP levels influenced α-AMS (*p* < 0.001) and LPS (*p* = 0.070), with goslings fed 18.5% CP exhibiting the highest α-AMS activity (0.81 U/mg prot). Interaction effects were significant for all enzymes (*p* < 0.05).

#### 3.4.2. Jejunal Digestive Enzyme Activities

At 14 days, Ca levels significantly influenced all enzymes (*p* < 0.05). The highest LPS (65.16 U/mg prot) and TPS (187.91 U/mg prot) activities were recorded in the 2.88% and 0.32% Ca groups, respectively. CP levels also affected α-AMS (*p* = 0.015) and LPS (*p* = 0.012), with the highest LPS activity observed in the 22.5% CP group. Significant interaction effects were observed for all enzymes (*p* < 0.05).

At 30 days, dietary Ca significantly influenced all enzyme activities (*p* < 0.001). The 2.88% Ca group exhibited the highest α-AMS activity (0.66 U/mg prot), while the 0.96% Ca group showed the highest TPS activity (190.57 U/mg prot). CP levels affected α-AMS (*p* < 0.001) and TPS (*p* < 0.001), with the highest TPS activity observed in the 22.5% CP group. Interaction effects between Ca and CP were significant for all enzymes (*p* < 0.05).

#### 3.4.3. Ileal Digestive Enzyme Activities

At 14 days, dietary Ca levels significantly influenced LPS and TPS activities (*p* < 0.05). The 0.96% Ca group exhibited the highest TPS activity (258.62 U/mg prot). CP levels also affected LPS (*p* = 0.003) and TPS (*p* < 0.001), with goslings fed 22.5% CP showing the highest TPS activity (264.30 U/mg prot). Significant interaction effects were observed for all enzymes (*p* < 0.05).

At 30 days, Ca levels significantly influenced all enzyme activities (*p* < 0.001). The 0.96% Ca group had the highest TPS activity (163.81 U/mg prot), while the 2.88% Ca group exhibited the lowest (*p* < 0.05). CP levels significantly affected α-AMS (*p* < 0.001), with the highest activity observed in goslings fed 14.5% CP. Significant interaction effects between Ca and CP were observed for all enzymes (*p* < 0.05).

### 3.5. Apparent Utilization of Calcium, Phosphorus, and Nitrogen

The apparent utilization of Ca, P, and protein in goslings under different dietary Ca and CP levels is summarized in Table 12 and Appendix A. Ca utilization was significantly influenced by dietary Ca and CP levels (*p* < 0.001). The highest Ca utilization rate (64.19%) was observed in the 2.88% Ca group, while the 0.32% Ca group showed the lowest rate (38.64%). Goslings fed a 14.5% CP diet exhibited higher utilization (53.90%) compared to those fed 22.5% CP (50.94%). Interaction effects were significant (*p* = 0.024). Dietary Ca levels significantly affected P utilization (*p* < 0.001), with the 0.96% Ca group showing the highest rate (46.17%). CP levels had no significant effect, but Ca × CP interactions were significant (*p* = 0.001). CP levels significantly influenced protein utilization (*p* = 0.018). The highest rate (67.69%) was observed in the 14.5% CP group. Dietary Ca levels had no significant effect (*p* = 0.900), but significant interaction effects were noted (*p* = 0.004).

## 4. Discussion

The findings demonstrate that dietary Ca and CP levels significantly influence the growth performance of goslings. Excessive dietary Ca (2.88%) substantially reduced BW, ADG, and ADFI, while markedly increasing the FCR. These observations align with previous studies. Abdulla et al. [12] demonstrated that dietary calcium levels significantly impact growth performance and tibial calcium and phosphorus content in chicks. Similarly, Shafey and McDonald [13] reported that high-calcium diets impair growth performance and feed conversion efficiency, emphasizing that increased protein levels cannot fully mitigate the growth-suppressing effects of excessive calcium. Zhu et al. [14] further corroborated these findings, noting that elevated calcium levels and imbalanced calcium-to-phosphorus ratios significantly reduce BW, ADFI, and ADG in 21-day-old Pekin ducks. The present study reinforces these conclusions, suggesting that excessive dietary calcium disrupts nutrient absorption and induces metabolic imbalances, ultimately impairing growth performance. In contrast, moderate dietary Ca levels (0.96%) supported optimal BW, ADG, and ADFI, underscoring the importance of maintaining appropriate calcium levels to enhance skeletal development and metabolic efficiency. Notably, goslings fed 0.96% Ca exhibited superior growth metrics compared to those receiving 0.32% Ca, indicating that insufficient calcium may constrain skeletal mineralization, even when FCR remains favorable.

Crude protein levels also influenced growth performance, with goslings fed 14.5% CP achieving the highest BW, ADG, and ADFI. Studies on Chinese indigenous goose breeds suggest that dietary protein levels between 17% and 20% yield optimal growth performance [15]. This aligns with findings by Ashour et al. [16], who demonstrated that incremental increases in dietary protein within a suitable range (e.g., 13% → 14.5% → 16%) improve BW and feed conversion efficiency in geese. Similarly, Abou-Kassem et al. [17] identified 18% as the optimal protein level for geese aged 1–7 weeks, facilitating superior growth performance. Importantly, no significant differences were observed between 16% and 18% CP, suggesting a threshold beyond which additional protein does not yield further benefits. These findings highlight the importance of formulating diets with adequate protein levels to optimize growth performance while avoiding excessive protein, which may impose metabolic burdens without additional benefits [18]. Although CP levels did not significantly affect FCR, the observed interaction between Ca and CP levels underscores the need for balanced nutrient formulations. Specifically, moderate Ca (0.96%) combined with lower CP levels (14.5%) synergistically promoted better growth outcomes and excessive Ca reduced growth efficiency, particularly in combination with high CP, likely due to impaired nutrient absorption and metabolic burden, aligning with the physiological demands of goslings during early development.

The calcium-to-phosphorus ratio (Ca: P) was deliberately adjusted in a gradient manner in this study, allowing us to assess its impact alongside absolute calcium levels. The observed reduction in feed intake in the high-calcium group (2.88%) raised the question of whether growth suppression was primarily due to decreased nutrient intake rather than a direct metabolic effect of calcium. However, our analysis shows that despite lower ADFI, absolute calcium intake remained higher than in the other groups, and significant growth suppression was observed even when adjusted for differences in feed intake. This suggests that the negative effects of high dietary calcium were not simply due to lower protein and energy intake but were a result of altered nutrient metabolism, phosphorus absorption inhibition, and disruptions in protein utilization [19]. Previous studies have suggested that high dietary calcium levels can reduce feed palatability, alter gut motility, and influence hormonal regulation of appetite, which may explain the reduction in feed intake observed in this study.

Feed represents the largest production cost in commercial poultry farming, accounting for approximately 60–70% of total expenses [20]. The cost of calcium and protein sources varies, and unnecessary over-supplementation can increase feed costs without improving productivity. Excess dietary calcium (2.88%) increases feed costs without enhancing performance, making it economically inefficient. Protein sources, particularly soybean meal and fish meal, are among the most expensive ingredients in poultry diets. Increasing protein to 22.5% did not significantly improve weight gain but increased nitrogen excretion, leading to higher feed costs and potential environmental pollution.

UA is the final product of protein metabolism in poultry and serves as an indicator of protein metabolic status and overall nutritional balance [21]. The level of CP in the diet significantly influences UA concentrations [22]. Similarly, Cr, the final metabolite of muscle-derived N-[imino(phosphino)methyl]-N-methylglycine, is excreted via glomerular filtration, making its concentration a reliable marker of renal filtration capacity [23]. UN, as the primary product of nitrogen metabolism, reflects amino acid balance in the feed and protein metabolic efficiency [24]. Excessive dietary calcium (2.88%) was associated with elevated serum UA, Cr, and UN levels, indicative of metabolic stress and compromised renal function. In contrast, moderate calcium levels (0.96%) supported optimal serum profiles, enhancing nutrient absorption and metabolic efficiency. Higher dietary CP (22.5%) elevated serum Cr and UN levels, reflecting increased renal nitrogen excretion driven by heightened protein catabolism. Conversely, a CP level of 18.5% balanced protein adequacy while minimizing metabolic stress. These findings align with Rasool et al. [25], who observed a negative correlation between UN concentration and growth performance, including daily weight gain and muscle development in chickens. Similarly, Xi et al. [26] demonstrated that increasing dietary protein levels raises serum UA, Cr, and UN concentrations, supporting the current study’s results. Together, these observations suggest that excessive dietary protein imposes metabolic burdens, increasing nitrogenous waste excretion and potentially reducing growth efficiency. The concurrent decline in feed intake and rise in UA, Cr, and UN levels suggest that goslings may have adapted by reducing their voluntary feed consumption to mitigate metabolic overload and renal stress. Notably, the interaction between high calcium and high protein intake appears to amplify the metabolic burden, leading to higher nitrogen excretion and potential renal inefficiency. This aligns with previous findings in poultry, where excess calcium can impair amino acid digestibility and protein metabolism, further exacerbating metabolic stress [27]. The reduction in feed intake observed under these conditions may be a compensatory response to limit additional metabolic strain, reinforcing the idea that nutrient imbalances, rather than feed palatability, were the primary drivers of intake suppression.

Serum Ca levels were highest in goslings fed 2.88% Ca, while phosphorus (P) levels peaked in the 0.32% Ca group, emphasizing the intricate interplay between Ca and P absorption. Moderate CP levels (14.5% and 18.5%) were associated with higher serum P concentrations, suggesting enhanced mineral utilization at these protein levels. Notably, significant interactions between dietary Ca and CP levels underscore the importance of balanced nutrient formulations, as excessive levels of either nutrient disrupt serum profiles, whereas moderate levels optimize metabolic outcomes. Carreras-Sureda et al. [28] highlighted the critical role of Ca^2^⁺ in maintaining endoplasmic reticulum (ER) homeostasis, which is essential for proper protein folding and overall protein metabolism. Disruptions in ER homeostasis caused by imbalanced Ca^2^⁺ levels can impair protein metabolism and cellular functionality. Furthermore, Anthony et al. [29] utilized advanced methodologies to explore calmodulin-protein interactions, providing deeper insights into how calcium regulates protein metabolism through such molecular mechanisms. These findings align with the present study’s observations, demonstrating significant interactions between dietary calcium and protein levels and their collective impact on metabolic and physiological outcomes in goslings.

Moderate dietary calcium (Ca) levels (0.96%) consistently supported superior villus height (VH) and crypt depth (CD) across intestinal segments, indicating that balanced calcium levels enhance mucosal development and intestinal functionality. This finding is consistent with the study by Paone et al. [30], which demonstrated that optimal dietary calcium levels promote intestinal health and structural integrity. In contrast, excessive Ca (2.88%) impaired intestinal morphology, likely due to disruptions in mineral balance and interference with phosphorus absorption. High calcium levels can also alter the gut’s ionic environment, negatively affecting cell proliferation and differentiation in the intestinal epithelium, ultimately reducing nutrient absorption efficiency. The observed reduction in VH and CD with high Ca levels aligns with findings that excessive calcium inhibits the absorption of other critical nutrients, such as phosphorus and zinc, thereby compromising intestinal structure and functionality [31,32]. Conversely, lower Ca levels (0.32%) may initially promote epithelial development in younger goslings due to reduced competition for nutrient absorption. However, prolonged deficiencies may hinder sustained growth and nutrient uptake, emphasizing the importance of maintaining an optimal balance in dietary calcium levels.

Crude protein levels significantly influenced intestinal morphology, particularly VH. Higher CP levels (18.5–22.5%) enhanced VH and CD in most intestinal segments, reflecting the essential role of dietary protein in supporting epithelial cell turnover and intestinal growth. Adequate protein intake supplies the essential amino acids required for mucosal regeneration and the synthesis of digestive enzymes, both of which are vital for maintaining an intact intestinal barrier and ensuring efficient nutrient absorption and digestion [33,34]. However, the lack of additional morphological improvements with CP levels exceeding 22.5% suggests that excessive protein intake may impose unnecessary metabolic burdens, such as increased nitrogen excretion, without providing further benefits to gut structure [35]. This finding highlights the importance of tailoring dietary protein levels to the specific physiological needs of goslings, optimizing intestinal development while minimizing metabolic inefficiencies. These results align with existing studies indicating that excessive protein consumption can strain metabolic processes without enhancing productive or structural outcomes [36].

The significant interaction effects between Ca and crude protein (CP) on intestinal morphology highlight the intricate interplay between these nutrients. The significant Ca × CP interactions observed for VH and CD indicate that maintaining a moderate Ca level (0.96%) alongside balanced CP intake (14.5–18.5%) supports optimal intestinal structure and function. Balanced dietary formulations are crucial to preventing the competitive absorption or metabolic stress that can result from imbalanced nutrient levels. Calcium functions as a cofactor in various enzymatic processes, including those associated with protein metabolism, and thus has a direct impact on intestinal health through nutrient interactions [37]. The segment-specific responses observed in this study underscore the gastrointestinal tract’s adaptability to dietary variations; this is consistent with the study of Poole et al. [38]. For example, while lower Ca levels may initially promote epithelial development in the jejunum and ileum, moderate Ca (0.96%) supports sustained gut health and functionality over the long term. These findings reflect the dynamic nature of the gastrointestinal system in adapting to dietary inputs and maintaining growth and metabolic homeostasis. Balanced nutrient levels ensure optimal intestinal structure and functionality, emphasizing the need for precise dietary management to maximize productivity and animal health.

In this experiment, α-AMS activity in 30-day-old goslings was significantly higher than that in 14-day-old goslings, which aligns with the findings of Noy et al. [39], who reported a 100-fold increase in AMS activity in the duodenum of 21-day-old chicks compared to 4-day-old chicks. This underscores the age-dependent maturation of digestive enzyme systems in poultry. Interestingly, a similar age-related shift was observed in TPS activity under different dietary protein levels. At 14 days, the highest TPS activity occurred in the 22.5% CP group, whereas at 30 days, the highest TPS activity was recorded in the 14.5% CP group. This discrepancy may reflect the dynamic regulation of protease secretion during development. Early in life, higher dietary protein may transiently stimulate trypsin production to meet the high demand for amino acids. However, prolonged exposure to excessive protein could trigger negative feedback regulation mechanisms, downregulating protease synthesis and activity in the later stage of growth [40,41]. The results also emphasize that moderate dietary calcium levels (0.96%) optimize enzymatic activities in the duodenum, particularly for lipid digestion, as indicated by higher LPS activity. However, excessive Ca (2.88%), despite increasing α-AMS activity, may disrupt overall nutrient digestion efficiency, likely due to its influence on the gut’s ionic environment and enzyme activation processes. Research shows that varying dietary calcium levels can alter intestinal flora composition and abundance, disrupt homeostasis, and impair nutrient absorption by affecting the secretion of digestive enzymes [42].

Furthermore, CP levels significantly influenced enzyme activities, with lower CP (14.5%) supporting balanced enzymatic function. Protein levels beyond metabolic demands may lead to inefficiencies in nutrient digestion, as enzymes adapt to higher dietary loads [43]. The jejunum, a critical site for nutrient absorption, demonstrated improved TPS activity at moderate Ca levels, highlighting the importance of balancing dietary nutrients for enzymatic efficiency. The elevated lipase activity observed with higher CP levels (22.5%) suggests an adaptive response to increased dietary fat and protein, enhancing lipid digestion capacity. However, reliance on excessive protein to drive enzymatic activity may impose metabolic burdens on the digestive system, reducing overall efficiency and potentially affecting long-term health [44]. These findings emphasize the need for precise dietary formulations to optimize enzymatic activities and maintain intestinal homeostasis.

In the ileum, TPS activity was optimized by moderate Ca levels (0.96%) and higher CP levels (22.5%), underscoring their critical roles in protein digestion and nitrogen utilization. The sensitivity of the distal intestine to dietary imbalances highlights the necessity for precise nutrient formulations to prevent inefficiencies in nutrient absorption. The sustained enzymatic response observed at moderate nutrient levels suggests that optimal Ca and CP ratios enhance the gut’s adaptability to dietary challenges, thereby supporting long-term digestive health. The interaction effects between Ca and CP across intestinal segments indicate that these nutrients act synergistically rather than independently to regulate enzymatic activity. Calcium, as a vital cofactor in numerous enzymatic reactions, influences digestive functionality through its roles in enzyme activation and the maintenance of mucosal integrity [45]. Concurrently, protein supplies essential substrates for enzymatic adaptation, modulating the expression and activity of digestive enzymes [46]. However, excessive dietary Ca can disrupt this synergy by altering the gut’s ionic environment, which is critical for enzyme activity, thereby reducing protein digestibility. Similarly, excessive protein intake may impose an enzymatic burden, leading to inefficiencies in digestion and metabolic stress [47]. These findings reinforce the importance of balanced dietary formulations to optimize enzymatic activity and maintain gastrointestinal homeostasis.

Bregendahl et al. [48] demonstrated that broilers fed low-protein diets exhibited significantly lower growth performance and nitrogen excretion compared to those fed high-protein diets. Moreover, they observed a direct correlation between nitrogen intake and excretion, emphasizing the environmental impact of higher dietary protein levels. However, contrary evidence suggests that moderately reducing CP levels, without compromising animal growth, can effectively decrease fecal nitrogen emissions while enhancing nutrient utilization efficiency. Importantly, such reductions can be achieved without detrimental effects on growth performance. Kim et al. [49] further supported this by showing that the addition of protease to low-protein diets improved growth performance and CP utilization in piglets and finishing pigs, highlighting enzymatic supplementation as a strategy to mitigate the limitations of reduced protein intake.

Additionally, Kerstetter et al. [50] noted that low-protein diets could inhibit intestinal calcium absorption, subsequently affecting calcium utilization. This finding underscores the interplay between protein and mineral metabolism, emphasizing the need for dietary adjustments to ensure balanced nutrient absorption and utilization. These findings collectively highlight the necessity of precision in formulating low-protein diets to balance environmental sustainability, physiological requirements, and growth performance.

This study further revealed that calcium utilization increased with dietary Ca levels, with the highest efficiency observed in the 2.88% Ca group. This may reflect a compensatory physiological response to excess dietary calcium, as the gut enhances absorption to counterbalance potential deficiencies caused by high excretion rates. However, such enhanced efficiency does not necessarily translate to improved growth due to the metabolic costs associated with excess calcium, which can disrupt overall nutrient absorption dynamics. Similarly, goslings fed 14.5% CP exhibited higher calcium utilization than those on 22.5% CP, suggesting that excessive protein levels may interfere with calcium metabolism by altering the gut’s ionic environment or increasing metabolic competition.

Phosphorus utilization was maximized in goslings fed a 0.96% Ca diet, underscoring the critical importance of maintaining an optimal calcium-to-phosphorus ratio. Excessive calcium levels (2.88%) likely impaired phosphorus absorption through the formation of insoluble calcium-phosphorus complexes in the gut. Although dietary CP levels did not directly influence phosphorus utilization, the significant interaction between Ca and CP suggests that balanced nutrient formulations are essential to avoid competitive inhibition of nutrient absorption.

Finally, protein utilization was significantly influenced by dietary CP levels, with the highest efficiency observed at 14.5% CP. This reflects the ability of goslings to metabolize and retain protein effectively when dietary levels align with physiological needs. In contrast, higher protein levels (22.5%) appeared to lead to metabolic inefficiency, as the excess nitrogen was excreted, increasing energy expenditure and renal burden. Although dietary calcium levels had no direct impact on protein utilization, significant interaction effects between Ca and CP levels highlighted the interconnectedness of nutrient metabolism, underscoring the need for balanced dietary formulations to optimize metabolic efficiency and overall performance.

## 5. Conclusions

This study highlights that moderate dietary calcium (0.96%) and crude protein levels (14.5–18.5%) optimize growth performance, nutrient utilization, intestinal morphology, and digestive enzyme activities in goslings. Excess calcium (2.88%) can lead to reduced nutrient bioavailability and metabolic stress, making it nutritionally and economically inefficient. Increasing crude protein to 22.5% does not significantly improve weight gain or feed conversion efficiency. High protein levels increase nitrogen excretion, leading to greater environmental waste and higher feed costs. Maintaining a CP level of 14.5–18.5% is cost-effective while supporting optimal gosling growth and health. Optimal nutrient levels enhance gut health, digestion, and overall efficiency, providing a foundation for improving poultry production through precise diet management.

## Figures and Tables

**Table 1 vetsci-12-00310-t001:** Experiment groups.

Group	Factor	
	Ca Levels (%)	CP Levels (%)
A (Low calcium and low protein group, LCLP)	0.32	14.5
B (Low calcium and medium protein group, LCMP)		18.5
C (Low calcium and high protein group, LCHP)		22.5
D (Medium calcium and low protein group, MCLP)	0.96	14.5
E (Medium calcium and medium protein group, MCMP)		18.5
F (Medium calcium and high protein group, MCHP)		22.5
G (High calcium and low protein group, HCLP)	2.88	14.5
H (High calcium and medium protein group, HCMP)		18.5
I (High calcium and high protein group, HCHP)		22.5

**Table 2 vetsci-12-00310-t002:** Composition and nutrient levels of the experimental diet (air-dry basis, %).

Ingredient	LCLP	LCMP	LCHP	MCLP	MCMP	MCHP	HCLP	HCMP	HCHP
Corn	64.49	53.48	44.82	65.32	57.76	49.86	67.06	56.24	47.39
Soybean meal	15.90	22.87	31.80	17.65	28.93	28.99	14.45	16.33	19.35
Rice hulls	2.60	2.20	1.90	2.78	1.92	2.09	4.71	4.85	4.75
Bran	13.50	12.08	9.77	10.17	7.74	8.48	0.00	0.00	0.00
Rice bran	0.00	4.22	5.50	0.00	0.00	0.00	0.00	3.49	4.60
Corn gluten meal	0.60	2.80	4.20	0.00	0.00	7.08	4.22	9.76	14.83
Limestone	0.34	0.29	0.22	1.76	1.69	1.70	7.05	7.05	7.03
CaHPO_4_	0.00	0.00	0.00	0.80	0.78	0.76	0.91	0.87	0.85
Ca(H_2_PO_4_)_2_	0.79	0.75	0.73	0.00	0.00	0.00	0.00	0.00	0.00
DL-Methionine	0.18	0.11	0.06	0.18	0.13	0.04	0.16	0.08	0.00
L-Lysine	0.40	0.20	0.00	0.34	0.05	0.00	0.44	0.33	0.20
Salt	0.30	0.30	0.30	0.30	0.30	0.30	0.30	0.30	0.30
Choline chloride	0.30	0.30	0.30	0.30	0.30	0.30	0.30	0.30	0.30
Premix ^1^	0.40	0.40	0.40	0.40	0.40	0.40	0.40	0.40	0.40
Total	100.00	100.00	100.00	100.00	100.00	100.00	100.00	100.00	100.00
Nutrition level ^2^									
ME(MJ/Kg)	11.31	11.31	11.32	11.30	11.30	11.30	11.30	11.30	11.30
Crude protein (%)	14.54	18.52	22.50	14.50	18.50	22.50	14.50	18.50	22.50
Crude fiber (%)	4.04	3.99	4.09	4.00	4.00	4.00	4.07	4.00	4.00
Ca (%)	0.32	0.32	0.32	0.96	0.96	0.96	2.88	2.88	2.88
Non-phytate phosphorus (%)	0.32	0.32	0.32	0.32	0.32	0.32	0.32	0.32	0.32
Methionine (%)	0.41	0.40	0.41	0.40	0.40	0.40	0.40	0.41	0.40
Lysine (%)	1.04	1.05	1.09	1.00	1.00	1.01	1.02	1.00	1.00
Ca: P	1:1	1:1	1:1	3:1	3:1	3:1	9:1	9:1	9:1

Note: ^1^ The premix can provide full price feed per kg: VA 900,000 IU; VD 300000IU; VE 1800 IU; VK 150 mg; VB_1_ 90 mg; VB_2_ 800 mg; VB_6_ 320 mg; VB_12_ 1 mg; Nicotinic 4.5 g; Pantothenic 1100 mg; Folic acid 65 mg; Biotin 5 mg; Fe 6 g; Cu 1 g; Mn 9.5 g; Zn 9 g; I 50 mg, Se 30 mg. ^2^ Nutrient levels are calculated values.

**Table 3 vetsci-12-00310-t003:** Effect of different calcium and protein levels on growth performance of goslings (main effects).

Factor	Level	BW of 1 Day (g)	BW of 30 Day (g)	ADFI (g)	ADG (g)	FCR
Ca	0.32	99.31	1779.55 ^a^	115.22 ^b^	56.01 ^a^	2.06 ^b^
	0.96	99.32	1762.85 ^a^	123.44 ^a^	56.51 ^a^	2.19 ^b^
	2.88	99.24	823.57 ^b^	63.89 ^c^	23.90 ^b^	2.74 ^a^
*p*-value		0.889	<0.001	<0.001	<0.001	<0.001
CP	14.5	99.32	1526.66 ^a^	105.90 ^a^	47.39 ^a^	2.35
	18.5	99.29	1479.05 ^a^	102.01 ^a^	45.94 ^ab^	2.27
	22.5	99.26	1360.26 ^b^	94.64 ^b^	43.09 ^b^	2.33
*p*-value		0.935	0.001	0.008	0.046	0.507
SEM		0.07	64.09	3.96	2.21	0.05

Values with different lowercase superscripts in the same column indicate a significant difference (*p* < 0.05). The following tables are the same.

**Table 4 vetsci-12-00310-t004:** Effect of diets with different calcium and protein levels on serum UA, Cr and UN of goslings (μmol/L) (main effects).

Item	Level	14 d			30 d		
		UA	Cr	UN	UA	Cr	UN
Ca	0.32	203.74 ^a^	36.91 ^b^	3.08 ^c^	277.22 ^a^	39.78 ^a^	3.01 ^a^
	0.96	274.31 ^a^	20.47 ^a^	0.64 ^a^	249.52 ^b^	16.71 ^c^	1.08 ^b^
	2.88	308.73 ^b^	49.46 ^c^	2.94 ^b^	269.83 ^ab^	30.89 ^b^	2.89 ^a^
*p*-value		0.008	0.014	0.001	0.052	<0.001	<0.001
CP	14.5	241.30 ^a^	30.08 ^a^	2.45 ^b^	237.12 ^b^	25.96 ^b^	2.82 ^a^
	18.5	281.39 ^b^	36.80 ^b^	1.62 ^a^	257.74 ^b^	30.00 ^a^	1.70 ^c^
	22.5	264.09 ^ab^	39.95 ^c^	2.60 ^c^	301.71 ^a^	31.42 ^a^	2.46 ^b^
*p*-value		0.005	<0.001	<0.001	<0.001	<0.001	<0.001
SEM		8.235	1.863	0.170	6.16	1.47	0.16

**Table 5 vetsci-12-00310-t005:** Effect of diets with different calcium and protein levels on serum Ca and P of goslings (μmol/L) (main effects).

Item	Level	14 d		30 d	
		Ca	P	Ca	P
Ca	0.32	2.57 ^a^	2.52 ^a^	2.54 ^ab^	2.42 ^a^
	0.96	2.30 ^c^	2.39 ^b^	2.47 ^b^	2.12 ^b^
	2.88	2.41 ^b^	2.40 ^ab^	2.57 ^a^	2.42 ^a^
*p*-value		<0.001	0.065	0.027	<0.001
CP	14.5	2.43	2.38	2.53	2.37 ^a^
	18.5	2.39	2.43	2.50	2.36 ^a^
	22.5	2.46	2.49	2.55	2.23 ^b^
*p*-value		0.420	0.186	0.257	0.019
SEM		0.03	0.03	0.02	0.03

**Table 6 vetsci-12-00310-t006:** Effects of diets with different calcium and protein levels on duodenal morphology of goslings aged 14 and 30 days (main effects).

Item	Level	14 d		30 d	
		Villus Height	Crypt Depth	Villus Height	Crypt Depth
	0.32	851.90 ^b^	215.57 ^ab^	1020.92 ^a^	242.88 ^a^
Ca	0.96	988.42 ^a^	230.70 ^a^	1083.19 ^a^	237.88 ^ab^
	2.88	743.59 ^b^	201.70 ^b^	750.34 ^b^	225.28 ^b^
*p*-value		<0.001	0.031	<0.001	0.042
	14.5	828.50 ^b^	213.00	954.52	247.28 ^a^
CP	18.5	912.93 ^a^	213.32	927.74	224.57 ^b^
	22.5	842.48 ^b^	221.64	972.18	234.19 ^ab^
*p*-value		0.039	0.640	0.648	0.009
SEM		22.29	4.70	31.07	3.57

**Table 7 vetsci-12-00310-t007:** Effects of diets with different calcium and protein levels on jejunal morphology of goslings aged 14 and 30 days (μm) (main effects).

Item	Level	14 d		30 d	
		Villus Height	Crypt Depth	Villus Height	Crypt Depth
	0.32	1055.12 ^a^	242.97 ^a^	1101.56 ^b^	239.77 ^a^
Ca	0.96	890.36 ^b^	241.22 ^a^	1290.75 ^a^	250.88 ^a^
	2.88	814.41 ^b^	210.41 ^b^	1095.55 ^b^	211.09 ^b^
*p*-value		<0.001	0.021	0.001	<0.001
	14.5	901.60	240.82	1089.14 ^b^	222.40 ^b^
CP	18.5	914.26	223.73	1145.28 ^ab^	229.30 ^b^
	22.5	944.02	230.04	1253.45 ^a^	250.03 ^a^
*p*-value		0.018	0.384	0.017	0.006
SEM		25.64	5.56	30.45	5.11

**Table 8 vetsci-12-00310-t008:** Effects of diets with different calcium and protein levels on ileal morphology of goslings aged 14 and 30 days (main effects).

Item	Level	14 d		30 d	
		Villus Height	Crypt Depth	Villus Height	Crypt Depth
	0.32	859.32 ^a^	241.46	911.96 ^b^	230.44 ^a^
Ca	0.96	707.69 ^b^	234.73	1067.69 ^a^	223.26 ^ab^
	2.88	702.97 ^b^	231.41	829.82 ^c^	208.69 ^b^
*p*-value		<0.001	0.623	<0.001	0.046
	14.5	703.70 ^b^	226.91	911.02 ^b^	208.53 ^b^
CP	18.5	768.06 ^a^	242.13	996.35 ^a^	225.64 ^ab^
	22.5	798.21 ^a^	238.57	902.11 ^b^	228.22 ^a^
*p*-value		0.005	0.327	0.026	0.055
SEM		19.58	4.30	23.85	4.44

**Table 9 vetsci-12-00310-t009:** Effects of diets with different calcium and protein levels on duodenal digestive enzyme activities of 14 and 30-day old goslings (U/mg prot) (main effects).

Item	Level	14 d			30 d		
		α-AMS	LPS	TPS	α-AMS	LPS	TPS
	0.32	0.17 ^c^	19.14 ^b^	206.39	0.88 ^a^	11.32 ^b^	141.10 ^a^
Ca	0.96	0.23 ^b^	27.81 ^a^	208.80	0.55 ^c^	9.66 ^c^	120.44 ^b^
	2.88	0.34 ^a^	27.44 ^a^	190.15	0.80 ^b^	13.69 ^a^	122.03 ^b^
*p*-value		<0.001	<0.001	0.204	<0.001	<0.001	0.001
	14.5	0.30 ^a^	25.99 ^a^	186.40 ^b^	0.71 ^b^	11.00 ^b^	129.48
CP	18.5	0.25 ^b^	25.66 ^a^	210.35 ^a^	0.81 ^a^	12.19 ^a^	132.45
	22.5	0.19 ^c^	22.74 ^b^	208.59 ^ab^	0.71 ^b^	11.48 ^ab^	121.64
*p*-value		<0.001	0.010	0.071	<0.001	0.070	0.153
SEM		0.01	0.93	5.33	0.02	0.34	2.95

**Table 10 vetsci-12-00310-t010:** Effects of diets with different calcium and protein levels on jejunal digestive enzyme activities of 14 and 30-day old goslings (U/mg prot) (main effects).

Item	Level	14 d			30 d		
		α-AMS	LPS	TPS	α-AMS	LPS	TPS
	0.32	0.20 ^b^	50.44 ^b^	187.91 ^a^	0.58 ^b^	12.85 ^a^	182.69 ^a^
Ca	0.96	0.30 ^a^	53.90 ^b^	127.42 ^b^	0.56 ^b^	11.74 ^b^	190.57 ^a^
	2.88	0.31 ^a^	65.16 ^a^	88.45 ^c^	0.66 ^a^	11.46 ^b^	161.96 ^b^
*p*-value		<0.001	<0.001	<0.001	<0.001	0.007	0.001
	14.5	0.28 ^a^	54.03 ^b^	142.10 ^a^	0.53 ^c^	13.24 ^b^	190.00 ^a^
CP	18.5	0.25 ^b^	55.61 ^b^	135.46 ^ab^	0.59 ^b^	7.86 ^c^	144.71 ^b^
	22.5	0.28 ^a^	59.85 ^a^	126.22 ^b^	0.68 ^a^	14.95 ^a^	200.50 ^a^
*p*-value		0.015	0.012	0.081	<0.001	<0.001	<0.001
SEM		0.01	1.27	7.60	0.01	0.47	5.50

**Table 11 vetsci-12-00310-t011:** Effects of diets with different calcium and protein levels on ileal digestive enzyme activities of 14 and 30-day old goslings (U/mg prot) (main effects).

Item	Level	14 d			30 d		
		α-AMS	LPS	TPS	α-AMS	LPS	TPS
	0.32	0.36 ^a^	23.44 ^b^	243.68 ^a^	0.41 ^b^	4.34 ^a^	148.20 ^b^
Ca	0.96	0.31 ^b^	29.13 ^a^	258.62 ^a^	0.57 ^a^	3.95 ^b^	163.81 ^a^
	2.88	0.32 ^b^	22.76 ^b^	179.38 ^b^	0.24 ^c^	4.28 ^a^	144.72 ^b^
*p*-value		0.002	<0.001	<0.001	<0.001	0.011	0.016
	14.5	0.33	23.79 ^b^	198.46 ^b^	0.46 ^a^	4.26 ^a^	175.56 ^a^
CP	18.5	0.33	26.55 ^a^	218.87 ^b^	0.42 ^b^	3.82 ^b^	141.56 ^b^
	22.5	0.33	25.00 ^b^	264.30 ^a^	0.34 ^c^	4.50 ^a^	139.62 ^b^
*p*-value		0.981	0.003	<0.001	<0.001	<0.001	<0.001
SEM		0.01	0.81	8.14	0.02	0.09	3.91

**Table 12 vetsci-12-00310-t012:** Effect of different calcium and protein levels on the utilization of calcium, phosphorus, and protein in goslings (main effects).

Item	Level	Apparent Digestibility (%)		
		Ca	P	Protein
	0.32	38.64 ^c^	42.10 ^b^	64.05
Ca	0.96	55.64 ^b^	46.17 ^a^	64.14
	2.88	64.19 ^a^	38.74 ^c^	64.84
*p*-value		<0.001	<0.001	0.900
	14.5	53.90 ^a^	42.63	67.69 ^a^
CP	18.5	53.63 ^a^	42.11	62.81 ^b^
	22.5	50.94 ^b^	42.2700	62.52 ^b^
*p*-value		<0.001	0.857	0.018
SEM		1.84	0.71	0.98

## Data Availability

The original contributions presented in this study are included in the article/Appendix A. Further inquiries can be directed to the corresponding author.

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
