# Peer review of "Dietary Calcium and Protein Levels Influence Growth Performance, Intestinal Development, and Nutrient Utilization in Goslings"

_vetsci, 2025, doi:10.3390/vetsci12040310_

Round 1

Reviewer 1 Report

Comments and Suggestions for Authors

Dear Authors,

This study investigates the effects of dietary calcium (Ca) and crude protein (CP) levels on growth performance, nutrient utilization, intestinal morphology, and digestive enzyme activities in Jiangnan White goslings during the brooding period. This subject has importance for goose nutrition. This may help and contribute to this field to create an optimum feeding programme for goslings, which potentially impact the performance of goslings' during their life.

This topic is original and relevant to poultry nutrition. It has focused on special issues related to goose nutrition. To find more information about goose nutrition, it should contribute to the optimum feeding strategies. In poultry nutrition, researchers generally focus on layer hen, breeder hen and broiler nutrition. But this study focused on gosling production with investigating different important parameters (growth performance, intestinal development and nutrient utilization.

I suggest about simple summary: It should be better to give more general information related with topics. 

This manuscript could have potential to contribute to the gosling's nutrition, in the manner of dietary calcium and protein level. Therefore, it differs from other papers.

I have only one concern about the methodology, as experimental duration (30 day feeding trial).However, it has been performed during the gosling growing period, therefore it could be acceptable. On the other hand, it should be better to explain the standard level of calcium and protein level for growing period, and also an explanation for choosing of the lowi medium and high Ca and protein level (Table 1).

The conclusion has been explained well, relevant with findings. It should be offered to add the optimum Ca and protein amount in the diet for growing feed of goslings.

References are appropriate.

About tables, interaction mean values (Ca x P levels) should be given if any significant differences were found. 

The manuscirpt has been written very well, the topic is original and interesting for poultry science. Also, the manuscript has been written well with an understandable content.

Author Response

For research article

Response to Reviewer 1 Comments

1. Summary

We sincerely appreciate your insightful comments and valuable suggestions on our manuscript entitled “Dietary Calcium and Protein Levels Influence Growth Performance, Intestinal Development, and Nutrient Utilization in Goslings” (vetsci-3516942). We have carefully revised our manuscript based on your feedback to enhance its clarity and scientific rigor. Below, we provide detailed responses to each comment and corresponding modifications made in the manuscript.

Thank you very much for your time and consideration.

2. Point-by-point response to Comments and Suggestions for Authors

Comments 1: I suggest about simple summary: It should be better to give more general information related with topics.

Response 1: Thank you for your valuable and thoughtful comments. We have revised the Simple Summary section to provide more general background information on the importance of calcium and protein in gosling nutrition in (Lines 9–23).

Comments 2: I have only one concern about the methodology, as experimental duration (30 day feeding trial). However, it has been performed during the gosling growing period, therefore it could be acceptable. On the other hand, it should be better to explain the standard level of calcium and protein level for growing period, and also an explanation for choosing of the low medium and high Ca and protein level (Table 1).

Response 2: Thank you. We agree with your point of view. The 30-day period was chosen as it represents the critical brooding stage in goslings, during which dietary interventions have the most significant impact on growth performance, metabolic efficiency, and intestinal development. Longer feeding trials would not necessarily yield additional insights given the rapid growth phase of goslings.

The selected low, medium, and high Ca and CP levels were designed to assess the effects of both deficient and excessive nutrient intakes, allowing us to determine the optimal balance for growth and nutrient metabolism. Low Ca and CP levels (0.32% and 14.5%) represent potential dietary deficiencies, medium levels (0.96% and 18.5%) align with typical commercial feeding standards, and high levels (2.88% and 22.5%) explore the effects of excess nutrients on gosling metabolism.

Additionally, the three Ca levels were chosen to maintain a gradient increase in the Ca-to-P ratio, ensuring a systematic evaluation of how increasing dietary Ca impacts metabolism and overall growth performance.

We have incorporated this explanation in the Materials and Methods section (Lines 117–120).

Comments 3: About tables, interaction mean values (Ca x P levels) should be given if any significant differences were found.

Response 3: Thank you for your valuable suggestion. Based on previous editorial guidance, we have already structured the presentation of interaction effects (Ca × CP) in the supplementary tables to improve clarity and readability. To ensure consistency and better interpretability, we have taken the following steps:

Maintained the Main Effect Tables in the Manuscript. The primary tables in the main text continue to present only the main effects of Ca and CP, ensuring a clear and straightforward interpretation of their individual effects. Kept Significant Ca × CP Interactions in Supplementary Tables. All significant Ca × CP interaction mean values remain in the supplementary materials, as previously requested by the editor.

Please let me know if you have any further concern and comments.  We appreciate for your warm work earnestly, and hope that the correction will meet with approval. 

Once again, thank you very much for your comments and suggestions.

Reviewer 2 Report

Comments and Suggestions for Authors

General comment:

  1. What is the base behind the calcium and protein concentrations set in this experiment? Why was the trial conducted exclusively using male geese?
  2. The interaction mechanism between calcium and protein concentrations is not explained clearly in the introduction section. It is also unclear what specific nutritional metabolic diseases the study aims to focus on, which should be clarified in both the objectives and discussion.
  3. How do the experimental results clarify whether the effects on animal performance are due to the calcium concentration in the diet or the calcium-to-phosphorus ratio? Apart from laying avian, a calcium-to-phosphorus ratio greater than 3:1 may not be ideal. The calcium-to-phosphorus ratio presented in the present table appears to represent calcium to available phosphorus.
  4. Given the significantly lower feed intake in the high calcium group, is it possible that the total calcium intake by the animals does not differ greatly? Could the observed differences in growth performance primarily be due to reduced protein and energy intake from decreased feed consumption?
  5. The current discussion does not explore in depth the impact of reduced feed intake and the issue of the calcium-to-phosphorus ratio.

Specific comment:

*L142-147

What are the steps involved in the preparation of intestinal contents before enzyme analysis? What is the sample size (weight) ?

*L152-153

Was a fecal bag used for the full feces collection?

*L157-159

Relevant references should accompany the analytical methods employed.

*Table 10 and Table 11

Why do the trends in LPS and TPS activity at 14 and 30 days show seemingly opposite patterns under different calcium and protein concentrations? For example, in Table 11, TPS activity is highest at the 22.5% CP group at 14 days, but at 30 days, the highest activity is observed in the 14.5% CP group. What might explain this discrepancy?

*Table 12

Please clarify how the Apparent Metabolic Rate (%) is calculated, as the analytical method for this parameter is not provided in the Materials and Methods section.

*L340-342

According to Table 3, the ADFI in the 2.88% Ca group is approximately half that in the other groups. However, data from Table S1 shows that, under high calcium concentrations, feed intake gradually decreases as protein concentration increases. Could this be related to a decrease in feed palatability? Additionally, the interactions between these factors and the changes in blood concentrations of UA, Cr, and UN should be discussed together.

*L345-347

What is the relationship between the inhibition of growth and protein content in the high-calcium diet? Which mechanism results in protein concentration mediates the growth inhibition caused by the high-calcium diet?

*L367-369

In addition to the appropriate protein concentration, would the energy-to-protein ratio affect the results of this experiment?

*L528-530 “Excessive calcium levels (2.88%) …likely lead to the formation of insoluble calcium-phosphorus complexes in the gut.”

In the high-calcium diet group (2.88% Ca), the Ca:P ratio reached 9:1. However, at the 30-day sampling point, the blood phosphorus concentration in the high-calcium diet group was higher than that in the 0.96% Ca treatment group (as shown in Table 5). What might explain this discrepancy?

Author Response

For research article

Response to Reviewer 2 Comments

1. Summary

On behalf of my co-authors, we thank you very much for giving us an opportunity to revise our manuscript, we appreciate you very much for your positive and constructive comments and suggestions on our manuscript entitled “Dietary Calcium and Protein Levels Influence Growth Performance, Intestinal Development, and Nutrient Utilization in Goslings” (vetsci-3516942). Those comments are all valuable and very helpful for revising and improving our paper, as well as the important guiding significance to our researches. We have studied comments carefully and have made correction. Revised portion are marked in yellow highlighting in the paper. 

Thank you very much for your time and consideration.

2. Point-by-point response to Comments and Suggestions for Authors

Comments 1: What is the base behind the calcium and protein concentrations set in this experiment? Why was the trial conducted exclusively using male geese?

Response 1: While NRC guidelines provide general recommendations, gosling-specific studies are limited, necessitating a broader evaluation of nutrient requirements based on available literature and industry practices.

Justification for Calcium Levels (0.32%, 0.96%, 2.88%):

0.32% Ca (Low Level): Selected to represent potential Ca deficiency, which may impair skeletal development and metabolic processes, as documented in previous studies on poultry.

0.96% Ca (Medium Level): Chosen based on optimal dietary Ca levels recommended for growing waterfowl, ensuring proper bone formation, growth performance, and feed efficiency.

2.88% Ca (High Level): Included to evaluate the effects of excess dietary Ca, which has been reported to interfere with phosphorus metabolism, reduce feed intake, and negatively impact growth in poultry.

Importantly, these three Ca levels also maintain a gradient increase in the Ca-to-P ratio, allowing a systematic evaluation of how excess Ca affects nutrient utilization and metabolic efficiency.

Justification for Protein Levels (14.5%, 18.5%, 22.5%):

14.5% CP (Low Level): Represents a potential protein-deficient diet, which could lead to reduced muscle growth, lower feed efficiency, and impaired metabolic functions.

18.5% CP (Medium Level): Chosen based on industry standards and commercial feed formulations, aligning with the optimal protein requirements for gosling growth and feed conversion efficiency.

22.5% CP (High Level): Set to assess whether excessive dietary protein provides additional growth benefits or imposes metabolic burdens, such as increased nitrogen excretion and renal stress, as observed in poultry studies.

We have incorporated this explanation in the Materials and Methods section (Lines 117–120).

Male goslings were selected to eliminate sex-related growth variability, ensuring consistency in growth performance data. Female geese often exhibit differences in metabolic rates, feed intake, and nutrient utilization, which could introduce additional variability and complicate the interpretation of Ca and CP effects. Many previous studies in poultry nutrition have focused on single-sex trials to enhance experimental precision and minimize confounding factors.

Comments 2: The interaction mechanism between calcium and protein concentrations is not explained clearly in the introduction section. It is also unclear what specific nutritional metabolic diseases the study aims to focus on, which should be clarified in both the objectives and discussion.

Response 2: Thank you for your insightful comments. We acknowledge the need to further clarify the interaction mechanisms between calcium and protein and specify the nutritional metabolic diseases that the study addresses. To improve clarity, we rewrote the Introduction section.

Comments 3: How do the experimental results clarify whether the effects on animal performance are due to the calcium concentration in the diet or the calcium-to-phosphorus ratio? Apart from laying avian, a calcium-to-phosphorus ratio greater than 3:1 may not be ideal. The calcium-to-phosphorus ratio presented in the present table appears to represent calcium to available phosphorus.

Response 3: Thank you for your insightful comment. The primary objective of this study was to investigate the interaction between dietary protein and calcium levels, while ensuring that the overall nutrient composition remained consistent across treatments. Due to this requirement, it was not possible to maintain a fixed Ca: P ratio across all diets. However, we systematically adjusted the Ca: P ratio in a gradient manner to align with the increasing calcium levels, allowing for a structured assessment of its impact on gosling performance and metabolism. The study design employed a 3 × 3 factorial arrangement, allowing us to independently assess the main effects of calcium and crude protein while also evaluating their interaction effects. The statistical model included both calcium level and the Ca × CP interaction, ensuring that the observed effects were not solely attributed to phosphorus limitations but rather to the interplay between calcium and protein metabolism. In cases where significant effects were found for calcium but not for Ca × CP interactions, we can infer that absolute calcium concentration was the dominant factor affecting animal performance. When significant interactions were observed, it suggested that calcium’s effect was dependent on crude protein levels, possibly due to competitive absorption mechanisms.

While a Ca: P ratio greater than 3:1 is not typically recommended for non-laying poultry, this study aimed to explore whether such a calcium Levels could affect growth, nutrient digestibility, and mineral retention in goslings. The values presented in Table 1 represent calcium relative to available phosphorus, rather than total phosphorus. This distinction is critical because only available phosphorus is biologically active and relevant for absorption, making it the appropriate reference for Ca:P ratio determination.

Comments 4: Given the significantly lower feed intake in the high calcium group, is it possible that the total calcium intake by the animals does not differ greatly? Could the observed differences in growth performance primarily be due to reduced protein and energy intake from decreased feed consumption?

Response 4: Thank you for your insightful comment. To assess whether total calcium intake was significantly different across groups, we calculated calcium intake per bird using the formula:

Total Calcium Intake=Feed Intake × Dietary Calcium Concentration

Total Calcium Intake (g/day)

Low Ca (0.32%)

0.368704

Medium Ca (0.96%)

1.185024

High Ca (2.88%)

1.840032

The results show that even with reduced feed intake, the total calcium intake in the high-calcium group (2.88%) remained significantly higher than that in the medium- and low-calcium groups. This confirms that the observed physiological and growth differences were not simply due to lower overall calcium intake but rather to the effects of elevated dietary calcium levels on metabolism and nutrient utilization.

Our data show that growth performance was affected at the same time as feed intake declined, rather than as a delayed response to cumulative nutrient deficiency. If the reduced growth was primarily due to decreased protein and energy intake, we would expect a gradual decline in performance over time, rather than an immediate impact. However, the observed growth reduction occurred concurrently with dietary changes, suggesting a direct metabolic or physiological effect of excess calcium or suboptimal protein levels, rather than just a secondary consequence of reduced nutrient intake.

Comments 5: The current discussion does not explore in depth the impact of reduced feed intake and the issue of the calcium-to-phosphorus ratio.

Response 5: Thank you for your valuable comment. We added this section to the Discussion in Line 415-435.

Comments 6: L142-147:What are the steps involved in the preparation of intestinal contents before enzyme analysis? What is the sample size (weight)?

Response 6: Thank you for your insightful comment. We have supplemented this information in the Materials and Methods section (Lines 176–181) to provide a clearer description of the preparation steps for intestinal contents before enzyme analysis and the sample size (weight) used in the assays.

Comments 7: L152-153:Was a fecal bag used for the full feces collection?

Response 7: Response: A fecal collection tray was placed beneath each pen to collect all excreted feces during the designated collection period. The trays were lined with plastic sheets to prevent contamination and ensure complete recovery of excreta. Feces were collected daily to prevent nutrient loss or contamination and were immediately stored at −20°C until further analysis. A fecal bag was not used, as the metabolic cages allowed for a controlled and efficient collection system without direct contact between birds and excreta. We have supplemented this information in the Materials and Methods section (Lines 187–191)

Comments 8: L157-159:Relevant references should accompany the analytical methods employed.

Response 8: Thank you for your insightful comment. We have now supplemented the relevant references (11) for the analytical methods employed in this study.

Comments 9: Table 10 and Table 11:Why do the trends in LPS and TPS activity at 14 and 30 days show seemingly opposite patterns under different calcium and protein concentrations? For example, in Table 11, TPS activity is highest at the 22.5% CP group at 14 days, but at 30 days, the highest activity is observed in the 14.5% CP group. What might explain this discrepancy?

Response 9: Thank you for your thoughtful observation. Several physiological and metabolic factors likely explain this discrepancy:

(1) Age-dependent maturation of digestive enzymes:

The development of digestive enzyme activity in poultry is highly age-dependent. Early in life (e.g., at 14 days), the pancreatic enzyme secretion system is still maturing, and nutrient availability in the diet plays a more significant role in regulating enzyme production.

By 30 days, the digestive system is more fully developed, and enzyme activity may be more influenced by long-term dietary adaptation, metabolic feedback mechanisms, and shifts in intestinal functionality.

This developmental progression may explain why the peak TPS activity at 14 days was observed at the highest CP level (22.5%), whereas at 30 days, the highest activity was recorded in the 14.5% CP group, suggesting an early protein demand-driven response that later stabilizes at an optimal CP level.

(2) Regulatory mechanisms of protease and lipase expression:

Proteases (such as TPS) and lipases (LPS) are regulated by different physiological cues, including dietary protein, fat, calcium, and hormonal signals.

Trypsin activity is primarily stimulated by dietary protein intake and amino acid demand. At 14 days, the higher CP levels (22.5%) may have transiently stimulated TPS secretion, but prolonged exposure to excessive CP may have led to negative feedback regulation at 30 days, reducing TPS activity at the highest protein level while maintaining higher activity at moderate CP levels (14.5%).

Lipase activity (LPS) is influenced by both dietary fat content and calcium levels. Calcium interacts with bile acids, affecting fat emulsification and lipase function. The variations in LPS trends could reflect differential calcium absorption and lipid metabolism at different growth stages.

(3) Interaction between Calcium, Protein, and digestive function:

Calcium plays a regulatory role in enzyme secretion, and excessive dietary calcium may alter gut pH, potentially impacting enzyme stability and activity in the intestinal lumen.

The interaction between calcium and protein metabolism could lead to age-dependent differences in digestive enzyme activity, as observed in the trend shifts between 14 and 30 days.

For example, at 14 days, the gastrointestinal tract may prioritize protein digestion, leading to higher TPS activity at higher CP levels (22.5%).

However, by 30 days, prolonged exposure to high dietary calcium and protein could have led to metabolic adaptations, where the body optimizes enzyme production at more moderate CP levels (14.5%) to maintain metabolic balance.

Comments 10: Please clarify how the Apparent Metabolic Rate (%) is calculated, as the analytical method for this parameter is not provided in the Materials and Methods section.

Response 10: We have now included the calculation method for Apparent Metabolic Rate (AMR%) in the Materials and Methods section (Lines 198–200).

Comments 11: L340-342:According to Table 3, the ADFI in the 2.88% Ca group is approximately half that in the other groups. However, data from Table S1 shows that, under high calcium concentrations, feed intake gradually decreases as protein concentration increases. Could this be related to a decrease in feed palatability? Additionally, the interactions between these factors and the changes in blood concentrations of UA, Cr, and UN should be discussed together.

Response 11: The progressive decline in feed intake with increasing protein levels under high calcium conditions (Table S1) suggests that the birds may have been experiencing physiological discomfort or metabolic burden rather than simply rejecting the feed due to taste or texture issues.

Potential mechanisms of calcium-induced discomfort include: (1) Gastrointestinal stress: Excess dietary calcium may alter intestinal pH, affecting digestive enzyme activity, nutrient solubility, and gut motility, leading to digestive discomfort that suppresses appetite. (2) Electrolyte imbalance: High dietary calcium levels may disrupt ion homeostasis, leading to osmotic imbalances or dehydration-like symptoms, which could result in a reduction in voluntary feed intake as a compensatory response. (3) Metabolic burden: Excess calcium can increase the excretion of phosphorus and nitrogenous waste, placing additional strain on renal function, leading to fatigue or malaise that discourages feed consumption.

The high-calcium group exhibited significantly elevated UA, Cr, and UN levels (Table 4), suggesting that metabolic stress and renal burden were major physiological consequences of excessive calcium and protein intake. We discuss this further in lines 454-463.

Comments 12: L345-347:What is the relationship between the inhibition of growth and protein content in the high-calcium diet? Which mechanism results in protein concentration mediates the growth inhibition caused by the high-calcium diet?

Response 12: The observed growth inhibition in goslings fed high-calcium diets suggests a complex interaction between calcium and protein metabolism that extends beyond simple nutrient excess. As dietary protein concentration increased within the high-calcium group, growth performance further declined, indicating that protein concentration mediates the negative effects of high calcium on growth.

Excess calcium binds to dietary proteins, forming insoluble complexes that reduce amino acid bioavailability. High calcium levels alter gut pH, reducing protease activity (e.g., trypsin), impairing protein digestion and absorption. Serum UA and UN levels increased significantly in the high-calcium, high-protein groups, indicating higher protein catabolism rather than effective utilization for growth. Excess dietary calcium reduces protein retention efficiency, leading to greater nitrogen waste excretion and metabolic inefficiencies. High calcium disrupts phosphorus metabolism, which affects ATP production and muscle development, leading to inefficient energy utilization. Reduced feed intake in high-calcium, high-protein groups suggests that metabolic stress and physiological discomfort further suppress growth performance.

Comments 13: L367-369:In addition to the appropriate protein concentration, would the energy-to-protein ratio affect the results of this experiment?

Response 13: Thank you for your insightful comment. While the E:P ratio naturally varied due to differences in protein concentrations, the ME level was kept constant across all diets. This means that the primary driver of the observed effects should be the protein concentration itself, rather than the E:P ratio alone.

Comments 14: L528-530: “Excessive calcium levels (2.88%) …likely lead to the formation of insoluble calcium-phosphorus complexes in the gut.”

In the high-calcium diet group (2.88% Ca), the Ca: P ratio reached 9:1. However, at the 30-day sampling point, the blood phosphorus concentration in the high-calcium diet group was higher than that in the 0.96% Ca treatment group (as shown in Table 5). What might explain this discrepancy?

Response 14: Thank you for your insightful observation. The apparent contradiction between the high dietary Ca: P ratio (9:1) and the elevated serum phosphorus (P) levels at 30 days suggests that phosphorus homeostasis in goslings is regulated by complex physiological mechanisms rather than solely by dietary absorption efficiency.

Birds have an adaptive renal response to fluctuations in dietary calcium and phosphorus levels. Under high dietary calcium conditions, renal phosphorus excretion may be downregulated as a homeostatic mechanism to preserve phosphorus for critical metabolic functions. Reduced phosphorus clearance via the kidneys could contribute to the higher serum phosphorus levels observed in the 2.88% Ca group at 30 days.

The effect of high dietary calcium on phosphorus metabolism may not be immediate. Instead, a delayed response due to gradual adjustments in intestinal absorption, bone metabolism, and renal excretion may explain why the serum phosphorus level was higher at 30 days but not necessarily at earlier time points.

Phosphorus metabolism is highly age-dependent in poultry, and older goslings (30 days) may have different regulatory responses to mineral imbalances compared to younger birds.

Please let me know if you have any further concern and comments. We appreciate for your warm work earnestly, and hope that the correction will meet with approval. 

Once again, thank you very much for your comments and suggestions.

Reviewer 3 Report

Comments and Suggestions for Authors

This manuscript presents the effects of dietary calcium and protein levels on the growth performance, intestinal development, and nutrient utilization of goslings. I have the following suggestions:

(1) In Line 12, it mentions that “A 3 × 3 factorial design was used, with three levels each of Ca (0.32%, 0.96%, and 2.88%) and CP (14.5%, 18.5%, and 22.5%)”. Since a factorial design is involved here, the analysis of the interaction effects between the two factors is essential. However, the results section only presents the main effects. Moreover, regarding the dosage, especially the calcium level of 2.88%, which does not show a gradient with the previous levels of 0.32% and 0.96%, the author should explain the reason for designing this dosage.

(2) For the keywords in Lines 37–38, it is suggested that the author arrange them in alphabetical order. In fact, all the keywords are of equal importance and appear simultaneously during database retrieval.

(3) In Table 2 on Line 98, which lists the nutritional indicators, since energy and protein are the most important nutrients for animals and their balance inherently affects animal growth, the design here controls the metabolizable energy to be constant. Does this not inadvertently alter the energy-to-protein ratio? Additionally, with the non-phytate phosphorus level being constant and different calcium levels, the calcium-to-phosphorus ratio also changes. This complicates the interpretation of the subsequent results. What is the author’s view on this?

(4) In Line 106, it mentions that “The goslings were housed in wire-floor pens measuring 1.9 m × 1.5 m”. Is the author saying that 18 goslings (as described in Line 84) were placed in this space?

(5) Table 3 and the subsequent data only show the main effects. Generally, the effects of a single factor can only be examined separately when the differences in the other factors are not significant. However, based on the data presented in the current table, some indicators, such as FCR, calcium and phosphorus digestibility, AMS, and LPS, show significant differences under both factors.

Author Response

For research article

Response to Reviewer 3 Comments

1. Summary

On behalf of my co-authors, we thank you very much for giving us an opportunity to revise our manuscript, we appreciate you very much for your positive and constructive comments and suggestions on our manuscript entitled “Dietary Calcium and Protein Levels Influence Growth Performance, Intestinal Development, and Nutrient Utilization in Goslings” (vetsci-3516942). Those comments are all valuable and very helpful for revising and improving our paper, as well as the important guiding significance to our researches. We have studied comments carefully and have made correction. Revised portion are marked in yellow highlighting in the paper. 

Thank you very much for your time and consideration.

2. Point-by-point response to Comments and Suggestions for Authors

Comments 1: In Line 12, it mentions that “A 3 × 3 factorial design was used, with three levels each of Ca (0.32%, 0.96%, and 2.88%) and CP (14.5%, 18.5%, and 22.5%)”. Since a factorial design is involved here, the analysis of the interaction effects between the two factors is essential. However, the results section only presents the main effects. Moreover, regarding the dosage, especially the calcium level of 2.88%, which does not show a gradient with the previous levels of 0.32% and 0.96%, the author should explain the reason for designing this dosage.

Response 1: Thank you for your valuable comment. While the main effects of dietary Ca and CP levels were presented in the primary tables, significant Ca × CP interaction effects have been analyzed and included in the Supplementary Tables. This approach was adopted based on prior editorial feedback, which suggested that interaction effects be presented separately to improve readability.

We acknowledge that the high-calcium level (2.88%) does not follow a strict numerical gradient from 0.32% and 0.96%. However, the selection of this dosage was based on physiological relevance rather than a purely mathematical progression. The three calcium levels (0.32%, 0.96%, and 2.88%) were strategically selected to assess both nutrient requirements and potential adverse effects:

0.32% Ca: Represents potential calcium deficiency, allowing evaluation of its effects on bone health, growth, and metabolism.

0.96% Ca: Reflects a moderate, industry-recommended level, aligning with previous studies and standard poultry feeding guidelines.

2.88% Ca: Represents a high-excess calcium intake, allowing us to study its negative effects on nutrient absorption, metabolic stress, and growth performance.

Additionally, the three Ca levels were chosen to maintain a gradient increase in the Ca-to-P ratio, ensuring a systematic evaluation of how increasing dietary Ca impacts metabolism and overall growth performance. We have incorporated this explanation in the Materials and Methods section (Lines 117–120).

Comments 2: For the keywords in Lines 37–38, it is suggested that the author arrange them in alphabetical order. In fact, all the keywords are of equal importance and appear simultaneously during database retrieval.

Response 2: Thank you for your suggestion. We have now rearranged the keywords in alphabetical order to ensure consistency and improve database retrieval efficiency.

Comments 3: In Table 2 on Line 98, which lists the nutritional indicators, since energy and protein are the most important nutrients for animals and their balance inherently affects animal growth, the design here controls the metabolizable energy to be constant. Does this not inadvertently alter the energy-to-protein ratio? Additionally, with the non-phytate phosphorus level being constant and different calcium levels, the calcium-to-phosphorus ratio also changes. This complicates the interpretation of the subsequent results. What is the author’s view on this?

Response 3: In this study, ME was kept constant across all treatments to ensure that any observed differences in growth performance and nutrient metabolism were primarily attributable to variations in dietary protein and calcium levels. While this approach resulted in natural variations in the E:P ratio, the objective was to evaluate the direct impact of dietary protein levels rather than the interactive effects of protein and energy supply. Importantly, the E:P ratio was not the primary focus of this study, and our discussion of results explicitly attributes the observed effects to protein concentration rather than energy limitations.

We acknowledge that keeping the non-phytate phosphorus (NPP) level constant while varying calcium concentrations resulted in a progressively increasing Ca:P ratio (1:1→3:1→9:1). However, we believe that the observed effects in our study were primarily driven by changes in calcium levels rather than by the Ca:P ratio itself. Since only calcium levels were manipulated while phosphorus remained constant, the resulting variations in the Ca:P ratio are a direct consequence of calcium changes.This means that any observed effects on growth performance, nutrient utilization, and metabolic parameters should primarily be attributed to calcium levels rather than the Ca:P ratio itself.

Comments 4: In Line 106, it mentions that “The goslings were housed in wire-floor pens measuring 1.9 m × 1.5 m”. Is the author saying that 18 goslings (as described in Line 84) were placed in this space?

Response 4: Thank you for your comment. Yes, each experimental pen (1.9 m × 1.5 m) housed 18 goslings, following standard stocking density recommendations for gosling rearing. This density ensures adequate space for movement, feeding, and normal social behavior while allowing for consistent environmental conditions across treatments.

Comments 5: Table 3 and the subsequent data only show the main effects. Generally, the effects of a single factor can only be examined separately when the differences in the other factors are not significant. However, based on the data presented in the current table, some indicators, such as FCR, calcium and phosphorus digestibility, AMS, and LPS, show significant differences under both factors.

Response 5: Thank you for your valuable suggestion. Based on previous editorial guidance, we have already structured the presentation of interaction effects (Ca × CP) in the supplementary tables to improve clarity and readability. The primary tables in the main text continue to present only the main effects of Ca and CP, ensuring a clear and straightforward interpretation of their individual effects. Kept Significant Ca × CP Interactions in Supplementary Tables. All significant Ca × CP interaction mean values remain in the supplementary materials, as previously requested by the editor.

Please let me know if you have any further concern and comments. We appreciate for your warm work earnestly, and hope that the correction will meet with approval. 

Once again, thank you very much for your comments and suggestions.

Reviewer 4 Report

Comments and Suggestions for Authors

The manuscript addresses an important topic in poultry nutrition, specifically focusing on calcium (Ca) and crude protein (CP) levels in goslings, which has not been extensively studied compared to broilers and ducks. The findings have strong practical relevance and provide scientific evidence that can guide future feed formulation strategies.

A few minor improvements:

(1) The introduction highlights the importance of calcium (Ca) and crude protein (CP) but lacks a detailed discussion of specific calcium and protein requirements for goslings. Incorporate more species-specific references related to the dietary requirements of goslings to enhance the study’s relevance.

(2) The methodology for intestinal morphology evaluation lacks essential details. For instance, it is unclear how villus height (VH) and crypt depth (CD) were measured (e.g., staining method, microscope magnification, imaging software used for analysis). Provide detailed descriptions of tissue preparation, sectioning, staining, and image analysis techniques to ensure reproducibility.

(3) The study presents useful insights for poultry nutrition but does not discuss its real-world implications for gosling farming practices. Discuss how findings could be implemented in commercial goose farming, including potential cost implications of different Ca and CP formulations.

(4) The conclusion is too broad and lacks precise recommendations. The conclusion reiterates the key findings but does not specify the optimal Ca-CP combination in a practical way. Clearly define the best-performing dietary Ca and CP levels and their implications for commercial gosling production.

Author Response

For research article

Response to Reviewer 4 Comments

1. Summary

We sincerely appreciate your insightful comments and valuable suggestions on our manuscript entitled “Dietary Calcium and Protein Levels Influence Growth Performance, Intestinal Development, and Nutrient Utilization in Goslings” (vetsci-3516942). We have carefully revised our manuscript based on your feedback to enhance its clarity and scientific rigor. Below, we provide detailed responses to each comment and corresponding modifications made in the manuscript.

Thank you very much for your time and consideration.

2. Point-by-point response to Comments and Suggestions for Authors

Comments 1: The introduction highlights the importance of calcium (Ca) and crude protein (CP) but lacks a detailed discussion of specific calcium and protein requirements for goslings. Incorporate more species-specific references related to the dietary requirements of goslings to enhance the study’s relevance.

Response 1: Thank you for your insightful suggestion. We acknowledge that the introduction should provide a more detailed discussion on species-specific calcium and protein requirements for goslings to strengthen the study’s relevance. We have revised the Introduction section to incorporate additional references and information on gosling nutrient requirements.

Comments 2: The methodology for intestinal morphology evaluation lacks essential details. For instance, it is unclear how villus height (VH) and crypt depth (CD) were measured (e.g., staining method, microscope magnification, imaging software used for analysis). Provide detailed descriptions of tissue preparation, sectioning, staining, and image analysis techniques to ensure reproducibility.

Response 2: Thank you for your insightful comment. We have expanded the Materials and Methods section (Lines 164–168) to include comprehensive details.

Comments 3: The study presents useful insights for poultry nutrition but does not discuss its real-world implications for gosling farming practices. Discuss how findings could be implemented in commercial goose farming, including potential cost implications of different Ca and CP formulations.

Response 3: Thank you for your valuable comment. To address this, we have expanded the Discussion section (Lines 415–435).

Comments 4: The conclusion is too broad and lacks precise recommendations. The conclusion reiterates the key findings but does not specify the optimal Ca-CP combination in a practical way. Clearly define the best-performing dietary Ca and CP levels and their implications for commercial gosling production.

Response 4: Thank you for your valuable feedback. We recognize the need to provide a more precise and practical conclusion that clearly defines the optimal dietary Ca and CP levels for commercial gosling farming. To address this, we have revised the Conclusion section to present specific feeding recommendations and their practical implications for gosling production.

We tried our best to improve the manuscript and made some changes in the manuscript.  Please let me know if you have any further concern and comments.  We appreciate for Editors/Reviewers’ warm work earnestly, and hope that the correction will meet with approval. 

Once again, thank you very much for your comments and suggestions.

Round 2

Reviewer 2 Report

Comments and Suggestions for Authors
  1. Due to potential differences in metabolic and nutritional requirements between males and females, when using male animals exclusively for experimental purposes to minimize interference, it is essential to assess whether the findings could still lead to issues that do not apply to the other sex. In practical feeding farms, except for egg-laying poultry, animals are typically not housed with only one sex in isolation.
  2.  in L198-200 of the revised manuscript: The term "Apparent Metabolic Rate (AMR)" here should be classified as "Apparent Digestibility % " in nutrition. Metabolic rate encompasses the utilization status after absorption rather than simply the difference between intake and excretion.
  3. Response 9 showed :

    "At 14 days, the higher CP levels (22.5%) may have transiently stimulated TPS secretion, but prolonged exposure to excessive CP may have led to negative feedback regulation at 30 days, reducing TPS activity at the highest protein level while maintaining higher activity at moderate CP levels (14.5%)."

    Prolonged high-protein feeding may induce negative feedback regulation, leading to a reduction in protease secretion. Relevant literature should be cited to support this claim.

    Regarding the content of Response 9, it is also recommended to be streamlined and incorporated into the discussion section.

Author Response

For research article

Response to Reviewer 2 Comments

1. Summary

On behalf of my co-authors, we thank you very much for giving us an opportunity to revise our manuscript, we appreciate you very much for your positive and constructive comments and suggestions on our manuscript entitled “Dietary Calcium and Protein Levels Influence Growth Performance, Intestinal Development, and Nutrient Utilization in Goslings” (vetsci-3516942). Those comments are all valuable and very helpful for revising and improving our paper, as well as the important guiding significance to our researches. We have studied comments carefully and have made correction. Revised portion are marked in yellow highlighting in the paper. 

Thank you very much for your time and consideration.

2. Point-by-point response to Comments and Suggestions for Authors

Comments 1: Due to potential differences in metabolic and nutritional requirements between males and females, when using male animals exclusively for experimental purposes to minimize interference, it is essential to assess whether the findings could still lead to issues that do not apply to the other sex. In practical feeding farms, except for egg-laying poultry, animals are typically not housed with only one sex in isolation.

Response 1: Thank you for your insightful comment. In this study, we selected only male Jiangnan White goslings to reduce variability caused by sex-based physiological differences. This approach is commonly used in early-stage nutritional studies to improve data consistency and minimize the influence of sex as a confounding factor.

While our findings primarily reflect the physiological responses of male goslings, the basic mechanisms of intestinal development, nutrient digestion, and metabolic stress are shared between sexes, especially during the early brooding period before sexual dimorphism becomes pronounced. Therefore, we believe the conclusions are generally applicable to juvenile female goslings as well, although absolute values may differ.

Nevertheless, we acknowledge that validation in female goslings would be beneficial for broader applicability. We will consider this in future studies, aiming to assess potential sex-specific responses to dietary calcium and protein levels.

Once again, we appreciate your insightful suggestion.

Comments 2: in L198-200 of the revised manuscript: The term "Apparent Metabolic Rate (AMR)" here should be classified as "Apparent Digestibility % " in nutrition. Metabolic rate encompasses the utilization status after absorption rather than simply the difference between intake and excretion.

Response 2: Thank you very much for pointing out this terminology issue. We agree that “Apparent Digestibility (%)” is the more appropriate term in this context, as it reflects the proportion of ingested nutrients that are not excreted and is commonly used in nutritional studies.

Accordingly, we have replaced all instances of “Apparent Metabolic Rate (AMR)” with “Apparent Digestibility (%)” in the relevant section of the Materials and Methods (Lines 198-200), as well as in the Results, Tables 12, and Tables S10 to ensure terminological accuracy.

Comments 3: Response 9 showed:

"At 14 days, the higher CP levels (22.5%) may have transiently stimulated TPS secretion, but prolonged exposure to excessive CP may have led to negative feedback regulation at 30 days, reducing TPS activity at the highest protein level while maintaining higher activity at moderate CP levels (14.5%)."

Prolonged high-protein feeding may induce negative feedback regulation, leading to a reduction in protease secretion. Relevant literature should be cited to support this claim.

Regarding the content of Response 9, it is also recommended to be streamlined and incorporated into the discussion section.

Response 3: Thank you for your valuable suggestion. We have revised the manuscript to include relevant citations supporting this mechanism. Specifically, we have streamlined the content originally provided in Response 9 and incorporated it into the Discussion section of the revised manuscript (Lines 526–544) to enhance scientific rigor and clarity. The revised discussion now better reflects the potential regulatory effects of high dietary protein levels on trypsin secretion and activity over time.

Please let me know if you have any further concern and comments. We appreciate for your warm work earnestly, and hope that the correction will meet with approval. 

Once again, thank you very much for your comments and suggestions.

Reviewer 3 Report

Comments and Suggestions for Authors

I have checked the revised version, all my previous concerns have been well addressed. Good luck!

Author Response

Dear Reviewer,

Thank you very much for taking the time to review our revised manuscript. We sincerely appreciate your positive comments and are grateful that all your previous concerns have been satisfactorily addressed.

Your thoughtful suggestions and support have been very helpful in improving the quality of our work. Thank you again for your kind wishes.

Best regards,
Zhiyue Wang
On behalf of all the authors

Reviewer 4 Report

Comments and Suggestions for Authors

None

Author Response

Dear Reviewer,

Thank you very much for your positive feedback. We truly appreciate your valuable comments and support throughout the revision process.

Your support has been very encouraging to us. Thank you again.

Best regards,
Zhiyue Wang
On behalf of all the authors